# High-Resolution Two-Dimensional Imaging of the 4H-SiC MOSFET Channel by Scanning Capacitance Microscopy

**DOI:** 10.3390/nano11061626

**Published:** 2021-06-21

**Authors:** Patrick Fiorenza, Mario S. Alessandrino, Beatrice Carbone, Alfio Russo, Fabrizio Roccaforte, Filippo Giannazzo

**Affiliations:** 1Consiglio Nazionale delle Ricerche—Istituto per la Microelettronica e Microsistemi (CNR-IMM), Strada VIII 5, 95121 Catania, Italy; fabrizio.roccaforte@imm.cnr.it (F.R.); filippo.giannazzo@imm.cnr.it (F.G.); 2STMicroelectronics, Stradale Primosole 50, 95121 Catania, Italy; santi.alessandrino@st.com (M.S.A.); beatrice.carbone@st.com (B.C.); alfio-lip.russo@st.com (A.R.)

**Keywords:** scanning probe microscopy, scanning capacitance microscopy, 4H-SiC, power-MOSFET

## Abstract

In this paper, a two-dimensional (2D) planar scanning capacitance microscopy (SCM) method is used to visualize with a high spatial resolution the channel region of large-area 4H-SiC power MOSFETs and estimate the homogeneity of the channel length over the whole device perimeter. The method enabled visualizing the fluctuations of the channel geometry occurring under different processing conditions. Moreover, the impact of the ion implantation parameters on the channel could be elucidated.

## 1. Introduction

Silicon-carbide (4H-SiC) metal-oxide-semiconductor field-effect transistors (MOSFETs) are raising the interest of the scientific community, owing to their applications and excellent performances in power electronics [1]. In the fabrication of vertical 4H-SiC MOSFETs, ion implantation is used to introduce dopant species (phosphorous for n-type and aluminum for p-type) in selective regions of the material, followed by high-temperature annealing for the electrical activation [2,3]. Hence, to accurately predict the device performance, both the active doping concentration and the geometry (e.g., size of the implanted region, junction depths, etc.) of the implanted MOSFET regions must be monitored at the nanoscale. In fact, while the diffusion coefficients in SiC are extremely low, the two-dimensional lateral spread of implanted atoms can affect the dopant distribution and, hence, the device behavior [4]. Hence, to accurately predict the MOSFET performance, both the active doping concentration and the geometry of the implanted regions (e.g., size, junction depths, etc.) must be monitored at the nanoscale. In fact, while the implantation doping in SiC is precisely localized after post-implantation annealing due to the extremely low diffusivity of the dopant species, the two-dimensional lateral spread of implanted atoms [4] and channeling effect in the hexagonal 4H-SiC lattice [5] can affect the dopant distribution.

In 4H-SiC power MOSFETs, the inversion channel length (in the order of few hundreds of nanometers) and the JFET (junction field-effect transistor) doping critically influence the threshold-voltage (V_th_), on-resistance (R_ON_), leakage current during the forward blocking mode, and gate-oxide-related ruggedness. Clearly, for high-current-level applications, 4H-SiC power MOSFETs are designed with large active areas (>10 mm^2^) and long perimeter (in the order of thousands of mm). Hence, the inversion of channel length and the JFET size must be extremely uniform along all the device perimeters to ensure the performance reproducibility. Thus, two-dimensional (2D) electrical imaging techniques combining high resolution (tens of nanometers) and the ability to probe large areas are needed to obtain statistically relevant information on the whole device periphery and, eventually, monitor anomalies of the electrical behavior.

In 4H-SiC MOSFET, the channel length is the distance of the p-type body from the n^+^-source junctions under the gate insulator [6,7]. These junctions’ positions depend on the doping of the n-type drift layer and on the electrical activation of aluminum and phosphorous implants employed for the formation of the body and source, respectively. Moreover, other factors can affect the device characteristics, e.g., the off-cut angle of the 4H-SiC crystals along the (11–20) direction, the shape of the hard masks used for selective ion implantation doping, etc. As an example, the lateral straggling of implanted Al in 4H-SiC has been observed to depend on the crystallographic orientation [8] and can result in asymmetric p-type doping profiles. Furthermore, for the degenerate phosphorous-implanted 4H-SiC, an electrical activation of n-type dopant in the order of 80% has been evaluated after annealing at typical temperatures of 1675 °C [9], whereas ~39% activation has been reported for high concentration Al implants (required for ohmic contact formation on the p-type body) under the same annealing conditions [10]. Clearly, the incomplete dopant activation in 4H-SiC introduces a degree of uncertainness for the device design. Finally, the fabrication steps may introduce topographic features and misalignments that can result in a non-uniform channel length over large distances. The electrical properties of p-type implanted layers after post-implantation annealing can be estimated by Hall measurements [10,11], which give an average behavior of “box-like” profiles. However, as the MOSFET body region is created by specific implants at different energies and doses, the knowledge of the active p-type dopant concentration depth profile is required, which cannot be easily assessed by averaged techniques. For this purpose, depth-resolved characterizations methods, such as SCM, can be applied to study the MOSFET body region [12], but they require specific epitaxial samples in order to calibrate the doping level.

Usually, scanning electron microscopy (SEM) analyses on cross-sectioned samples are routinely used to obtain information on the extension of the n^+^- and p^‒^-implanted regions in 4H-SiC power devices, by exploiting the sensitivity of secondary electrons’ contrast to potential variations in this wide-bandgap semiconductor [13]. However, this technique suffers from a certain degree of uncertainness in the determination of the electrical junction position, as it is sensitive only to high concentrations. 

In recent years, two-dimensional (2D) carrier profiling techniques based on atomic force microscopy, such as scanning capacitance microscopy (SCM) and scanning spreading resistance microscopy (SSRM), have been also explored to evaluate the electrically active profiles in ion-implanted 4H-SiC [14]. In particular, the SCM technique, based on local differential capacitance (dC/dV) measurements with a sliding metal tip, is very powerful for the delineation of the electrical junction position in semiconductor devices, by exploiting the sensitivity to the doping type of the dC/dV phase signal [15]. 

Usually, SCM is used for the semiconductor carrier profile and for the determination of the p–n junctions [16,17]. However, the cross-sectional methodologies (TEM, SEM, etc.) suffer from a lack of statistical relevance due to the fact that the information comes from a limited volume fraction of the device (~1 µm in depth) cross-section.

SCM analyses are often performed on cross-sections of 4H-SiC MOSFETs to evaluate the channel length [18]. However, cross-sectional analyses provide information only on a specific region of a device. On the other hand, 2D planar measurements are more adequate for monitoring the variations of the channel length along the device perimeter. 

In this paper, 2D scanning capacitance microscopy (SCM) in planar mode is used to monitor the channel length in 4H-SiC power MOSFETs with a high statistical relevance over areas in the order of 10^−2^ mm^2^. In particular, the method enabled the visualization of the fluctuations of the channel geometry occurring under different devices’ processing conditions. It is important to emphasize that in this planar configuration, standard SEM methods are not able to provide reliable information on the 4H-SiC power MOSFETs’ channel length. Moreover, the impact of the ion implantation parameters on the channel is discussed, pointing out the need for their fine tuning to optimize the trade-off between the total series resistance (R_on_) and threshold voltage (V_th_).

## 2. Materials and Methods

Vertical power MOSFETs were fabricated on 4°-off-axis n-type (0001) 4H-SiC epitaxial layers (between 9 × 10^15^ and 2 × 10^16^ cm^−3^), P-implanted source region (N_D_~10^20^ cm^−3^) and an Al-implanted body region (N_A_~10^17^ cm^−3^) [19]. The fabrication starts from zero-micropipe production-grade n^++^ substrates followed by CVD epitaxy process. After the growth of the epitaxial layer, the p-type body of the MOSFET is fabricated by ion implantation of aluminum. After a standard RCA cleaning, the gate oxide was a 40 nm thick deposited SiO_2_ layer [20]. Oxide layers have been deposited at a temperature higher than 700 °C by means of a low-pressure chemical vapor deposition (LPCVD) furnace using dichlorosilane (DCS) and nitrous oxide (NO) as silicon and oxygen precursors, respectively [21]. Nickel silicide is used to form ohmic contacts on the source body and drain. Polysilicon is used as a metal gate, and polyamide is used as surface passivation [22].

Some parameters of 4H-SiC MOSFETs concerning the doping of the epitaxial layer and the implantation of both the p-type body and the source region were varied, as reported in Table 1. In particular, a specific concentration is used as a reference for samples A and B. Devices C and D possess an increased body-doping concentration and a reduced epilayer-doping concentration, respectively. The device E possess both changed parameters (Table 1).

The power MOSFETs were tested by current–voltage (I–V) measurements, carried out using an Agilent B1505A parameter analyzer. After this macroscopic (i.e., device level) electrical characterization for the determination of the key electrical parameters (R_on_, V_th_), the devices were completely delayered from the passivation, metal, and gate oxide layers to expose the 4H-SiC bare surface. The delayering is obtained by dipping the device out of the package into an HF/H_2_O (40–60%) acid solution for 20 min [23]. Afterward, the delayered devices were subjected to an immersion in H_2_O_2_ at 40 vol. for 20 min [15]. This treatment also results in the formation of a native oxide on the SiC surface, which is necessary for the nanoscale resolution capacitance mapping by SCM [15]. These analyses were carried out using a DI3100 AFM with a Nanoscope V controller. Doped diamond-coated Si tips were employed to ensure electrical stability during large area scans on the structured 4H-SiC surface.

## 3. Results and Discussion

Figure 1a shows the basic structure of the planar power MOSFET in a cross-section where the channel regions are delimitated between the JFET and the source in the body-region ion implantation edges. On the other hand, Figure 1b shows the top view of the power MOSFET where the channel geometry is indicated, and, in particular, its length (L) is in the order of 200 nm and its width (W) is in the order of several millimeters.

As previously discussed, the common procedure to monitor the channel shape and length is in the cross-section by the device cleavage. Figure 2 shows the cross-sectional SCM image of a typical elementary cell of a 4H-SiC power MOSFET. In particular, the SCM image (Figure 2) also gives information on the space charge region (SCR). In particular, using the SCM, the SCR and the p-type doped body region results were distinguishable compared to the n-type JFET/drift region and to the source region. The channel can be recognized in the region near the surface delimited between the two n-type regions, and its shape can be influenced by several processing parameters.

In the proposed method, the information on the channel length is collected from the top of the device, thus enabling the visualization of the entire device perimeter, which is not possible by standard cross-section approach. Figure 3a shows the 3D schematic structure of the elementary cell of the MOSFET after the gate stack delayering. As can be noticed, the SCM tip can scan the different regions of the device, collecting information both on the channel length *L* and width *W*, important parameters to understand the physics of large area power MOSFETs. In order to better understand how important this aspect is, it is possible to consider how for a given technology design, the resulting electrical properties of the MOSFETs suffer from some degree of uncertainness on the *L* and *W* definition.

Figure 3b shows the AFM morphology map collected in contact mode onto the bare 4H-SiC semiconductor surface, exposing both the n- and p-type regions of the vertical MOSFET.

The in-plan top-view capacitance mapping [24] was performed while the semiconductor surface is scanned with the metal tip and a modulating bias with amplitude V at 100 kHz frequency is applied to the sample, and the capacitance variation ΔC in response to this modulation is recorded with the SCM sensor. Besides the |SCM| signal amplitude |ΔC|, which is related to the net active dopants concentration (N_A_-N_D_) in the semiconductor underneath the tip (Figure 3c), also the phase signal is recorded, which is very sensitive to the type of majority carriers in the region underneath the tip [25].

Figure 3b–d show, respectively, the morphology (AFM), SCM amplitude, and phase signal (ϕ) of a reference device. As can be seen in Figure 3d, the channel length is measurable and uniform (i.e., with a constant size) along the W direction in the un-optimized device. This information is undetectable using standard techniques.

The determination of the experimental and theoretical accuracy in delineating the electrical junction in p–n samples by SCM is still a challenging problem. This is due to the artifacts introduced by the sample cross-section preparation [26] such as morphological features (scratches) and surface contaminations that may introduce surface states, deteriorating the probed electrical signal. Furthermore, at a p–n junction, there exists a built-in depletion region where the net carrier concentration decreases from the bulk levels to zero on each side of the junction. In the p–n junctions, the measured SCM phase ϕ is positive for p-type material and negative for the n-type semiconductor. The apparent junction location (ϕ = 0°) can be moved throughout the depletion region in the vicinity of a p–n junction by SCM tip AC bias [27], as schematically depicted in Figure 4a. Hence, the SCM measurement may affect the estimation of the channel length increasing the depletion region under the tip. In order to overcome this problem, it is important to fix some experimental conditions. In order to avoid the perturbation of the depletion region in correspondence with the p–n junction, the DC bias is kept equal to zero. Hence, the best SCM setup is obtained by varying the AC bias amplitude. It has to be emphasized that the AC bias may also affect the depletion region in correspondence with the p–n junction. Then, it must be kept as small as possible, minimizing the noise/signal ratio. In this context, the chosen criterion is to keep a noise/signal ratio, minimizing the root mean square (RMS) of the SCM ϕ map. Figure 4b shows that the SCM RMS value decreases, increasing the AC tip bias. On the other hand, Figure 4b also shows how a given channel length is overestimated increasing the AC tip bias. Hence, the AC tip bias of choice is the value that lies at the minimum of the two lines depicted in Figure 4b (i.e., AC bias at 2 V).

After illustrating the in-plane SCM measurement configuration and the optimal bias conditions to evaluate the channel length, some applications of this method to different MOSFET devices will be illustrated.

Figure 5 is used to describe the procedure to extract the V_th_ value on the reference sample (A and B) and the characteristic MOSFETs curves used to measure the R_ON_ at a V_G_ = +15 V and at I_D_ value of 10 A. Hence, the typical values are R_ON_ = 23 mΩ (at a V_G_ = +15 V and I_D_ = 10 A), and the Vth (I_D_ = 250 µA) is about 2.7 V for the reference sample.

Figure 6 shows the empirical correlation between the resistance of a group of devices, expressed in terms of the MOSFET on-state resistance (R_On_) at a given current value (i.e., 10 A), as a function of the threshold voltage (V_th_) defined as the gate bias needed to turn on the MOSFET achieving a given current value (i.e., 250 µA). In order to better visualize the parameters under investigation, both the R_On_ and the V_th_ are normalized to the values obtained on sample A. Noteworthily, the experimental data are linearly correlated, i.e., the on-resistance increases with increasing the threshold voltage, in good agreement with the results presented by Noguchi et al. [28].

Figure 7a shows the SCM ϕ profile averaging 10 µm along the W width of the MOSFET channels under investigation. As can be seen, different channel lengths are measured on the devices under investigation. The channel lengths were measured, assuming as minimum the distance between the upper part of the positive SCM ϕ profile and as maximum the distance when the SCM ϕ crosses zero. From the interpolation between the experimental points and the crossing of SCM ϕ zero value, it can be argued that the spatial resolution is about 25 nm. The measured channel lengths are depicted as a function of the macroscopic parameters R_On_^Normalized^ and the V_th_^Normalized^ in Figure 7b,c, respectively. As can be noticed, a variation of the channel length smaller than 5% can produce appreciable variation of both R_On_ and V_th_, as in samples A and B, which are identical.

It can be concluded that the variation of the doping in the sample under investigation produced a variation in both the metallurgic position (i.e., the location where N_A_ = N_D_) and the space charge region (SCR) at the n^+^–p–n^−^ source–body–JFET junctions, as schematically represented in Figure 8a. Furthermore, the increase of the p-type body implantation dose in sample C produces an increased distance between the n^+^–p and p–n^−^ junctions, resulting in a wider channel (Figure 8b). In particular, for a fixed n^−^ epitaxial concentration in the JFET region, the p–n^−^ junction with the body is moved toward the n^−^ region once the p-type dose is increased (Figure 8b). By contrast, the n^+^–p junction between the source and the body is moved toward the source. On the other hand, the reduction of the n^−^ epitaxial layer concentration in sample D shifted the p–n^−^ junction toward the JFET and increased the SCR width between the body and the JFET (Figure 8c). Thus, the channel length of sample D is larger than A and B but smaller than C (Figure 8c). Finally, the combination of the increase of the body dose and a reduction of the epi-layer doping produced an intermediate channel length for sample E, as schematically depicted in Figure 8d.

In order to highlight the relevance of the proposed characterization technique, it is possible to consider how for a given technology design, the resulting electrical properties of the MOSFETs suffer from some degree of uncertainness of the *L* and *W* definition. In particular, an un-optimized lithographic process may induce large deviation from the ideal case. Figure 9a,b show respectively the morphology (AFM) and the SCM phase signal (ϕ) of an un-optimized device. As can be seen in Figure 9b, the channel length is not uniform along the W direction in the un-optimized device. This information is undetectable using standard cross-section techniques. In this specific case, the un-optimized device presented an anomalous R_On_ value for the designed V_th_ requirements, corresponding to a large on-resistance and sub-threshold leakage current.

## 4. Conclusions

In conclusion, a powerful 2D imaging method to visualize the channel in large perimeter 4H-SiC MOSFETs channel is presented using the SCM phase ϕ signal. The large areal statistical information (up to 10^−2^ mm^2^) can be used to validate the device processing and to obtain information on the device physics on semiconductors where the doping activation suffers from a certain degree of uncertainness. An appropriate sample choice is used to demonstrate the movement of the channel in the 3D MOSFET structure and to understand the electrical characteristics at the macroscopic scale.

## Figures and Tables

**Figure 1 nanomaterials-11-01626-f001:**
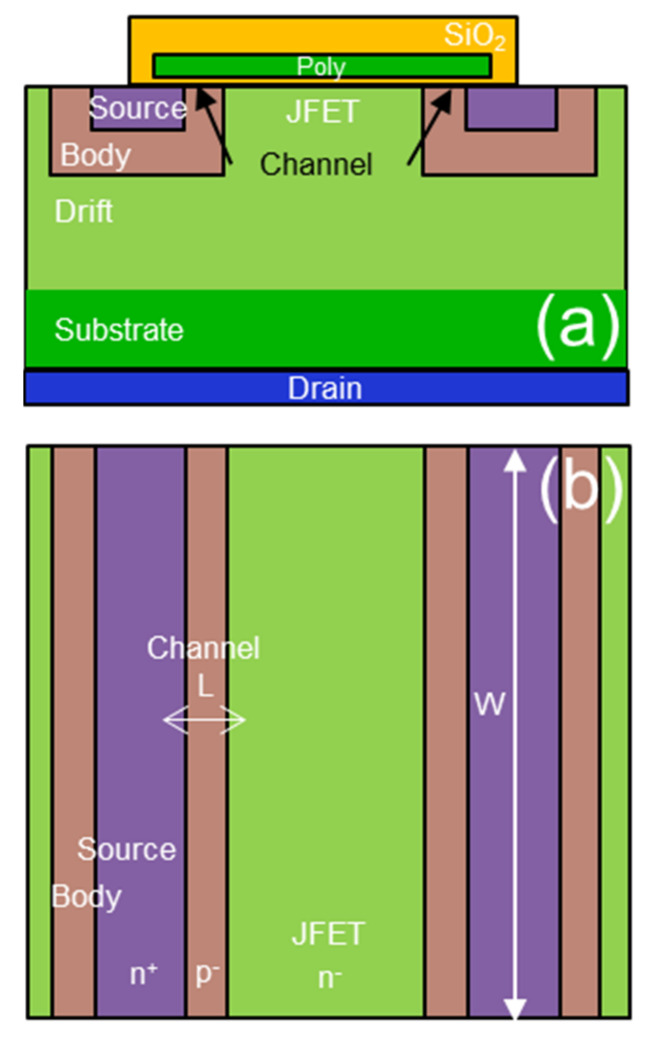
4H-SiC MOSFET schematic cross-section (**a**) and schematic top view (**b**).

**Figure 2 nanomaterials-11-01626-f002:**
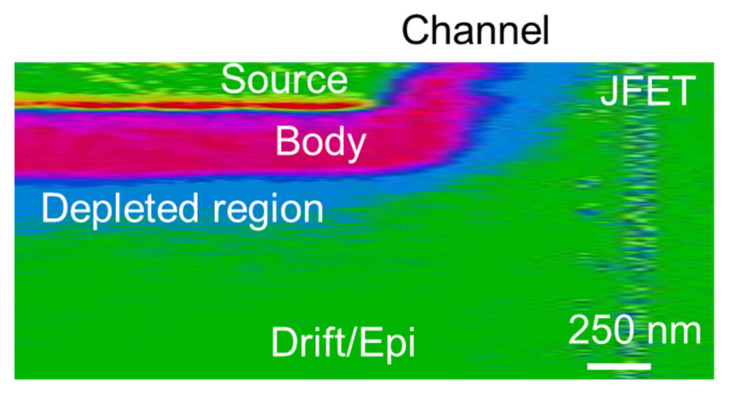
Cross section SCM image of the elementary MOSFET cell.

**Figure 3 nanomaterials-11-01626-f003:**
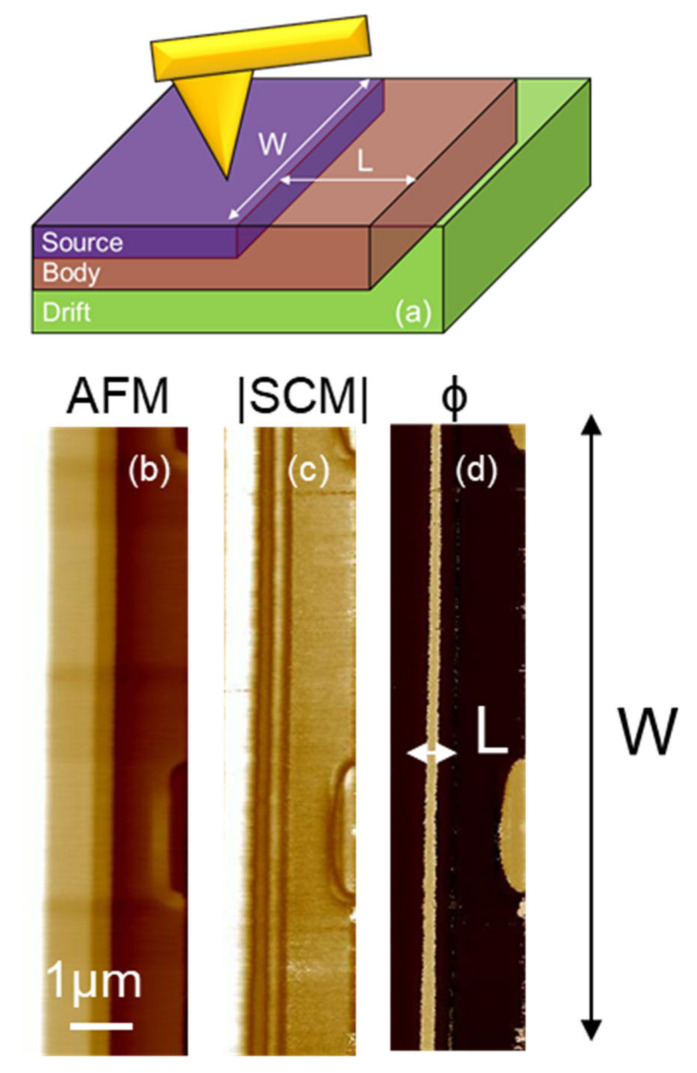
(**a**) Schematic description of the in-plan 2D SCM measurements on the MOSFET channel along W and across the L directions. (**b**) AFM morphology, (**c**) SCM amplitude, and (**d**) SCM phase (ϕ) of the reference sample.

**Figure 4 nanomaterials-11-01626-f004:**
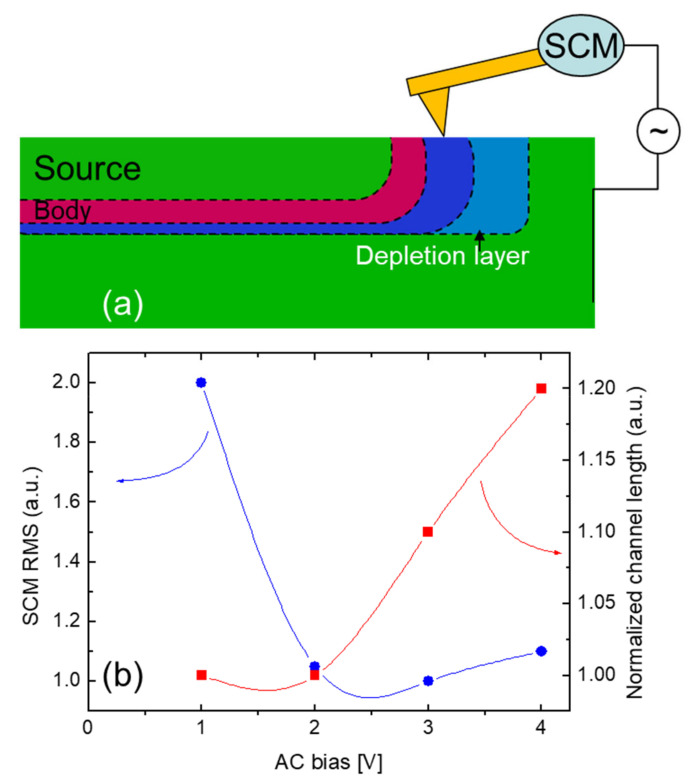
(**a**) Schematic description of the SCM tip influencing the depletion region at the p–n junction. (**b**) SCM RMS and normalized channel length vs. SCM tip AC bias.

**Figure 5 nanomaterials-11-01626-f005:**
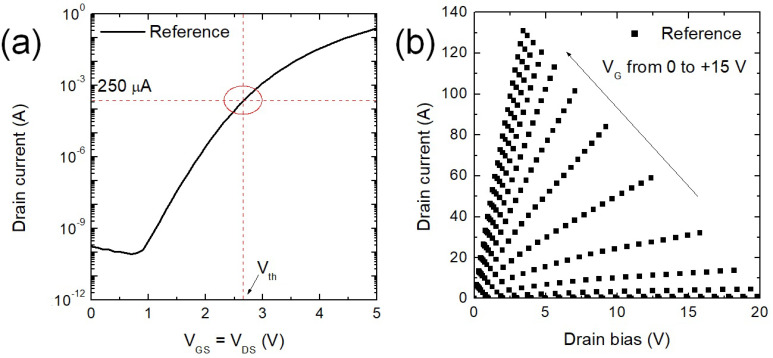
Experimental electrical measurements carried out on a reference MOSFET. The trans-characteristic I_D_-V_GS_ is shown to demonstrate how the V_th_ value is defined at fixed current value (**a**). (**b**) The output characteristics I_D_-V_D_ reported for V_G_ values up to +15 V are used to determine the R_ON_ value at 10 A.

**Figure 6 nanomaterials-11-01626-f006:**
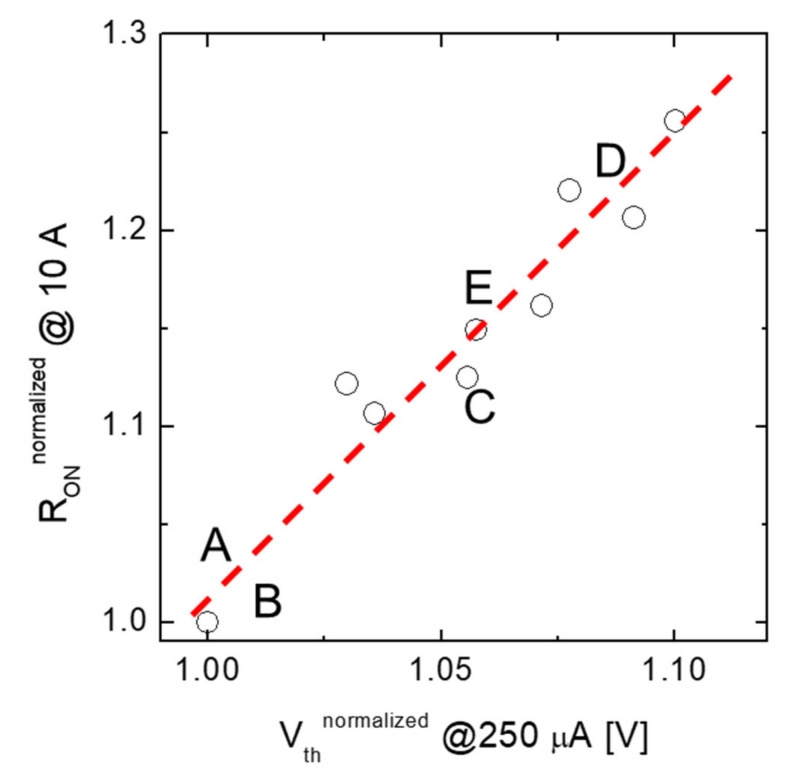
Empirical correlation between the on-resistance of a group of devices, R_On_ at 10 A vs. Vth (measured at 250 µA). Data are normalized to the values of sample A.

**Figure 7 nanomaterials-11-01626-f007:**
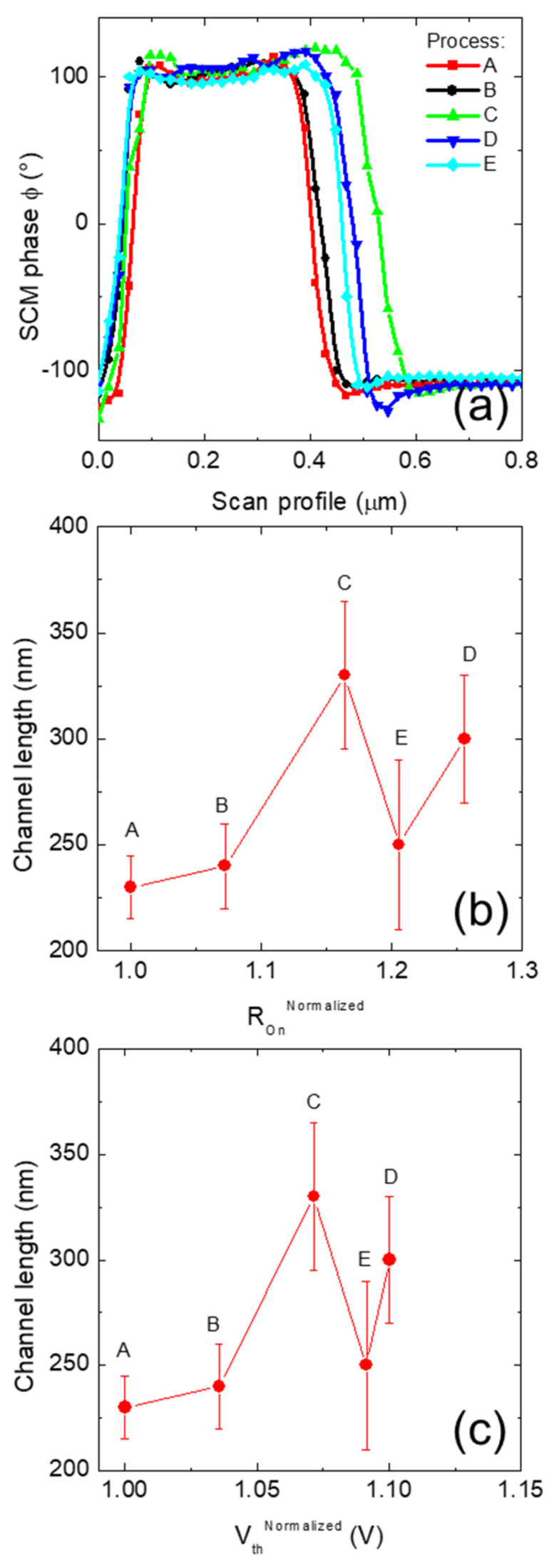
(**a**) SCM phase ϕ vs. scan profile averaged over 10 µm. Channel lengths vs. R_On_ (**b**) and vs. V_th_ (**c**).

**Figure 8 nanomaterials-11-01626-f008:**
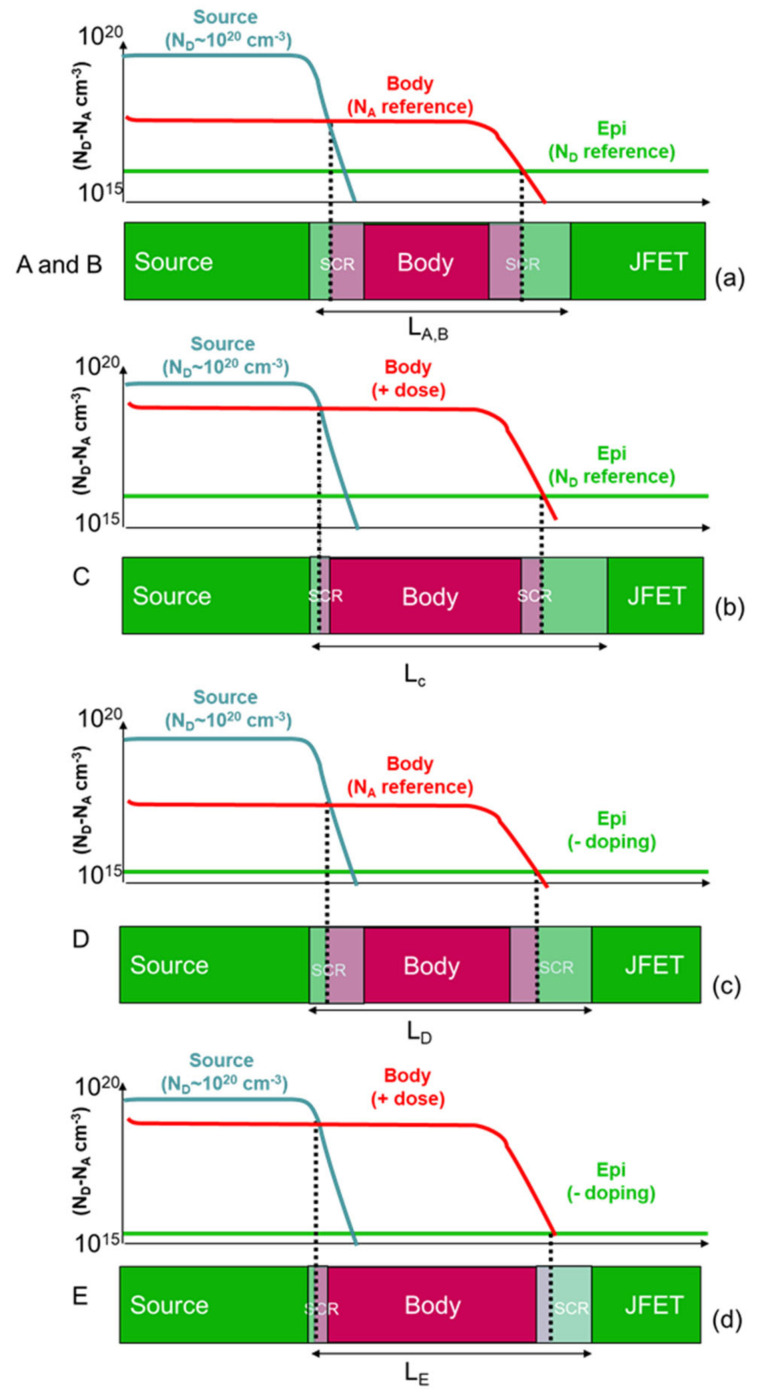
Schematic representation of the channel fabricated on the n^+^–p–n^−^ junction in the investigated samples. (**a**) Schematic representation of the reference samples (A and B), where the metallurgic junctions are represented by dashed lines when N_A_ = N_D_, generating the space charge regions (SCR) in the n- and p-type semiconductor according to the doping levels in the different samples. In particular, the increase of the body dose (**b**) or a reduction of the epi-layer doping (**c**) produced an increase of the channel lengths for MOSFET C and D, respectively. On the other hand, the combination of the increase of the body dose and a reduction of the epi-layer doping produced an intermediate channel length for sample E (**d**).

**Figure 9 nanomaterials-11-01626-f009:**
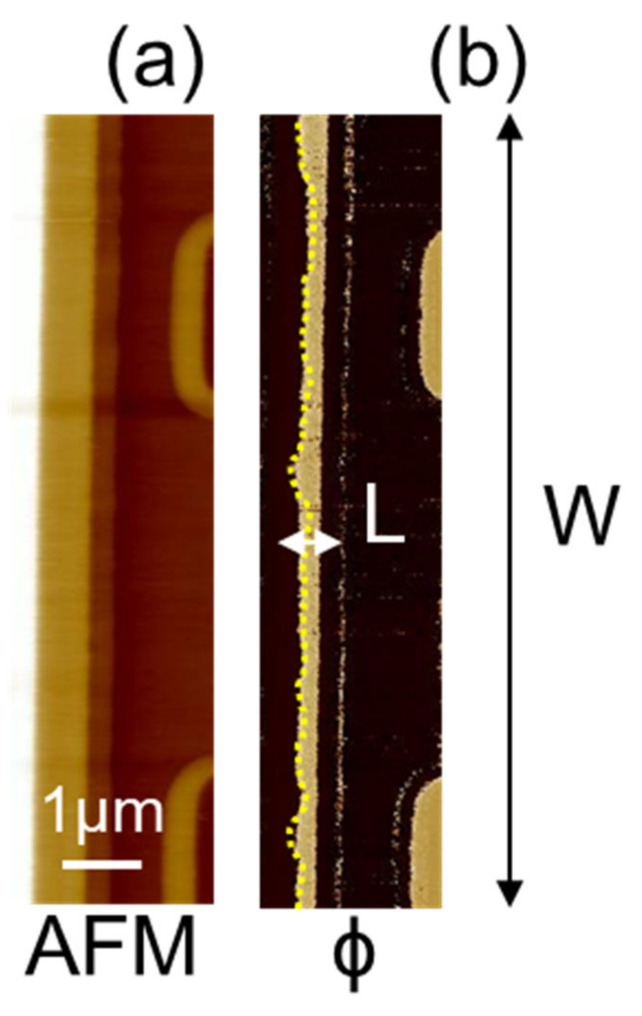
(**a**) AFM morphology and (**b**) SCM phase (ϕ) collected on a selected device that showed anomalous behavior compared with the desired design.

**Table 1 nanomaterials-11-01626-t001:** Sample description: two reference samples A and B are compared with a p-type implanted dose variation in C, epitaxial layer doping variation in D, and their combination in E.

Sample	Definition
A	Reference
B	Reference
C	+ Body dose
D	− Epi doping
E	− Epi doping + Body dose

## Data Availability

The data that support the findings of this study are available from the corresponding author upon reasonable request.

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
