# Peer review of "High-Resolution Two-Dimensional Imaging of the 4H-SiC MOSFET Channel by Scanning Capacitance Microscopy"

_nanomaterials, 2021, doi:10.3390/nano11061626_

Round 1
Reviewer 1 Report
The article summited by the authors has a real interest for the scientific community focused in actual SiC power devices and which try to understand the technological origin of the SiC MOSFET Vth drift.
Nevertheless the form of the presented article should be improved with some corrections that are proposed bellow.
1) Some remarks of the authors in the “Results and discussion” part can be presented earlier in the “Introduction” part and thus highlighting more the interest of their study and employed methods.
As for example the paragraph:
"Usually, SCM is used for the semiconductor carrier profile and for the determination of
the p-n junctions [17,18]. However, the cross-sectional methodologies (TEM, SEM, etc)
suffer of a lack of statistical relevance due to the fact that the information comes from a
limited volume fraction of the device (~ 1 μm in depth) cross section.”
2) In the “Introduction” part the authors state “In fact, while the diffusion coefficients in SiC are extremely low, the two dimensional lateral spread of implanted atoms can affect the dopant distribution and, hence, the device behaviour [4].”
In fact the “extremely low” doping diffusion in SiC is not a negatively feature. Compared to classic silicon technology an increasing in the accuracy definition and control of the sizes of FET channels can be obtained in silicon carbide as the doping is precisely localized in the ion implanted area without significant diffusion during the post-implantation annealing.
On the other hand, lateral spread depends on the crystal orientation with a canalization phenomenon which is difficult to control in hexagonal SiC as 4H-SiC.
3) “Clearly, the incomplete dopant activation in 4H-SiC introduces a degree of uncertainness for the device design."
The incomplete dopant activation in 4H-SiC can be calibrated with preliminary tests.
4) "In this paper, 2D scanning capacitance microscopy (SCM) in planar mode is used to
monitor the channel length in 4H-SiC power MOSFETs with a high statistical relevance
over areas in the orders of 10 4 μm 2"
As large power MOSFET bare dies are today commercialized by several suppliers, it is preferable to mention the scanning area in mm2 or even cm2, the classic unit used in microelectronics.
5) In the “Materials and Methods” part the authors mention;
"Vertical power MOSFETs were fabricated on 4°-off-axis n-type (0001) 4H-SiC epitaxial layers (in order of 1×10 16 cm -3 ),"
The doping concentration of the drift epitaxial layer should be known precisely in order to correctly estimate the channel and SCR lengths presented in this paper.
6) "The gate oxide was a 40 nm thick deposited SiO 2 layer [14]."
Is there any thermal oxide growth and specific cleaning before? Only deposition? In general poor quality of the SiO2/SiC interface is obtained in this case.
7) "the devices were completely delayered from the passivation, metal and gate oxide layers
to expose the 4H-SiC bare surface. The delayering is obtained by dipping the device out
of package in a HF/H 2 O (40%-60%) acid solution for 20 minutes [16]. This treatment also
results in the formation of a native oxide on the SiC surface, which is necessary for the
nanoscale resolution capacitance mapping by SCM."
In general completely remove of metalization from SiC is difficult to be achieved, in particular if typical Ni based alloys are formed for ohmic contacts with specific rapid thermal annealing.
Moreover, the authors should confirm and describe how a native oxide layer is obtained after an attack with relatively high concentration of HF. This last one is used to remove oxide layers from semiconductor surfaces!
8) In the “Results and discussion” part the authors mention:
"Fig. 5 shows the empirical correlation between the resistance of a group of devices –
expressed in terms of the MOSFET on-state resistance (R On ) at a given current value (i.e.
10 A) – as a function of the threshold voltage (V th ) defined as the gate bias needed to turn
on the MOSFET achieving a given current value (i.e. 250 μA). In order to better visualize
the parameters under investigation, both the R On and the V th are normalized to the values
obtained on the sample A."
The authors should mention also for the tested MOSFETs the real values of Ron and Vth
9) "(Fig. 7b). By contrast, the n + -p junction between the source and the body is moved toward the source due to its squeeze once
the p-type body concentration is increased."
The results presented in this figure are not clearly presented and often not consistent. By increasing the p-type doping, the SCR length can decrease close to the source.
The same evolution of the SCR length close to the drift epilayer should be obtained by increasing the doping of the p-type body from A/B samples to C sample and from D sample to E sample.
10) The paper seems to not be finished. No correlation between the calculations and the few results provided. Even if it was difficult to interpret without finding a correlation between the targeted dopings and the SCM images, presenting more results would have made it possible to show the interest of this technique in the direct mapping of SiC MOSFETs.
Author Response
Please have a look to the attached detailed pdf file

Reviewer 2 Report
This paper presents a 2D planar scanning capacitance microscopy (SCM) method to visualize with a high spatial resolution the channel region of large area 4H-SiC power MOSFETs and estimate the homogeneity of the channel length. Although the method is interesting and meaningful, some detailed information is missing.
- [page 2, line 84~92] Please show more information about the 4H-SiC MOSFETs, such as the source, drain, gate metal, passivation layer. The device fabrication process is not complete.
- [page 3, line 104~107] Here the devices were completely delayered from the passivation, metal and gate oxide layers to expose the 4H-SiC bare surface. The correspond SEM picture to check the 4H-SiC bare surface is need.
- [page 3, line 104~107] The delayered devices are used for the SCM mapping in the process. Why not use the material before device process to carry out the SCM mapping? If you would like to measure the DC performance, you can use two same materials: one for DC fabrication process and the other one is used for the SCM mapping. Please comment on it.
- [page 7, line 199~201] The related references for the definition of Ron and Vth are needed. Because the current is closed with the gate width in MOSFETs, the definition with current in the unit of A is not precise. Please comment on it. In addition, the related DC curves (ID-VD, ID-VG) are needed.
- [page 8, Figure 6] The SEM image of the device gate length is needed to confirm the gate length measured from the SCM.
Author Response

(The authors gave the same response as above.)
